# Ex vivo visualization of RNA polymerase III-specific gene activity with electron microscopy

Sina Manger[1], Utz H. Ermel[1] & Achilleas S. Frangakis [1✉]

The direct study of transcription or DNA–protein-binding events, requires imaging of individual genes at molecular resolution. Electron microscopy (EM) can show local detail of the genome. However, direct visualization and analysis of specific individual genes is currently not feasible as they cannot be unambiguously localized in the crowded, landmark-free environment of the nucleus. Here, we present a method for the genomic insertion of gene clusters that can be localized and imaged together with their associated protein complexes in the EM. The method uses CRISPR/Cas9 technology to incorporate several genes of interest near the 35S rRNA gene, which is a frequently occurring, easy-to-identify genomic locus within the nucleolus that can be used as a landmark in micrographs. As a proof of principle, we demonstrate the incorporation of the locus-native gene RDN5 and the locus-foreign gene HSX1. This led to a greater than 7-fold enrichment of RNA polymerase III (Pol III) complexes associated with the genes within the field of view, allowing for a significant increase in the analysis yield. This method thereby allows for the insertion and direct visualization of gene clusters for a range of analyses, such as changes in gene activity upon alteration of cellular or external factors.

[1] Buchmann Institute for Molecular Life Sciences and Institute for Biophysics, Goethe University Frankfurt, 60438 Frankfurt am Main, Germany.
✉email: achilleas.frangakis@biophysik.org

Electron microscopy (EM) allows for the direct visual analysis of cellular processes and the macromolecular complexes involved. A challenge for in vivo EM studies is the localization of a target structure within the crowded cellular environment. This can generally be overcome by using correlative light-electron microscopy (CLEM), in which a fluorophore-labeled target is first localized with (super-resolution) light microscopy and then imaged at high resolution with (cryo-) electron microscopy[1,2]. However, CLEM may not be useful when the target feature looks similar to its surroundings, as is the case for specific genes within the nucleus. Technologies such as DNA paint[3], the sequence-specific binding of labeled proteins[4], and invasive methods[5,6] that explore, e.g., electron-dense particles inserted in the cell do not allow for the localization and identification of specific genes. Thus, the reliable tracing of specific DNA segments and analysis of distinct sequences has not been possible up until now.

Here, we describe a method that enables the straightforward EM imaging of genetic events, such as transcription, chromatin modification or DNA–protein binding, in a native-chromatin environment within the nucleolus. The method relies on ribosomal DNA (rDNA) genes in *Saccharomyces cerevisiae*, which visually stand out from the surrounding chromatin and act as landmarks for the localization of specific genes that have been deliberately inserted for analysis. The genes for the four ribosomal RNAs (18S, 5.8S, 25S, and 5S rRNAs) are organized as multiple repeats on chromosome XII of *S. cerevisiae*. The standard repeat number is ~150, but there are strains with fewer repeats[7]. The high number of rDNA repeats within the genome facilitates multiple insertions of target genes and reduces the time necessary for finding them in the microscope. To increase the number of complexes that can be found, and thus the analysis yield, this approach incorporates multiple copies of the genes of interest, enabling them to be studied ex vivo using EM. As proof of concept, we incorporated multiple copies of two RNA Pol III-transcribed genes: the nucleolar locus-native *RDN5* (5 S rRNA) and the locus-foreign *HSX1* (arginine tRNA)[8].

## Results

**Strain engineering and analysis.** We used the stably Cas9-expressing, quadruple auxotrophic *S. cerevisiae* strain IMX672 (Y40595) (MATa ura3-52 trp1-289 leu2-3,112 his3Δ can1Δ::cas9-natNT2) along with a set of plasmids designed for cloning-free insertion of the CRISPR/Cas9 target sequence[9]. Standard CRISPR/Cas9 methods use templates for double-strand break repair that are designed to knock-in single genes[10]. Here, we aimed to incorporate multiple copies of the target genes into each of the 150 rDNA repeat loci on chromosome XII of the *S. cerevisiae* strain IMX672 in order to increase the yield for subsequent analysis (Fig. 1). Incorporating multiple gene copies with conventional methods would require very large templates produced by extensive cloning techniques[11]. Instead, we used overlapping oligonucleotides covering the target gene sequence together with adapter DNAs that were homologous to the junctions between genomic and repeat DNAs (donor DNAs), as previously suggested by Lancrey et al.[12] (Supplementary Table 2). The sequences of the oligomers used to assemble the 323 bp *RDN5* unit and the 284 bp *HSX1* unit were divided into 40–60 bp oligomers with an overlap of 18–20 bp, with the first and the last oligonucleotides overlapping as well (Fig. 1 and Supplementary Table 1). By screening different concentrations of the CRISPR RNA, the repeat-forming oligonucleotides, and the donor DNAs, as well as by modifying the length of the donor's homology arm, we adapted the protocol to enable incorporation at all 150 sites. The identity and position in the genome of the incorporated

DNA stretches with the *RDN5* and *HSX1* copies were confirmed by Sanger sequencing (Supplementary Fig. 1a, b).

We analyzed the number of inserted copies and the distribution of copy cluster length over the rDNA loci by Southern blotting of several transformants (Fig. 2a and Supplementary Fig. 2a; all raw and uncropped membranes of all Southern blots presented here can be found in Supplementary Fig. 6). An average of 5.3 *RDN5* copies and 2.3 *HSX1* copies was achieved per rDNA locus, with a maximum length of 16 *RDN5* copies and 7 *HSX1* copies over the whole genome (Fig. 2). By using a clone with empty incorporation sites (see first clone on the left in Supplementary Fig. 2a) as a parental strain in a subsequent experiment, we could iteratively increase the number of copies incorporated (Supplementary Fig. 2b). In the best-performing *RDN5* clone, 5x(O)*RDN5*, 780 *RDN5* copies were incorporated over 150 rDNA repeats, which amounts to approximately 252 kb of DNA. This massive manipulation of genomic integrity neither strongly impaired the cell growth nor the transcription of the rDNA repeats, as shown by the formation and preparation of physiological Miller trees (Supplementary Figs. 3 and 4). A statistical analysis of the incorporated gene copy numbers in relation to the different experimental incorporation conditions revealed that the 5x and 2.5x condition yielded the highest copy numbers for RDN5 and HSX1, respectively (Supplementary Fig. 5). We confirmed the 6-month stability of the genomic DNA by repeated Southern blot analysis of two edited clones after more than ten rounds of re-plating (Supplementary Fig. 2c).

**Analysis of *S. cerevisiae* spreads of edited clones by electron microscopy.** Next, we spread genomic DNA from the clones 5x(O)*RDN5* and 2.5x(J)*HSX1* on carbon-coated EM grids and imaged them in a transmission electron microscope (Figs. 2 and 3). DNA, polymerases, transcripts, and terminal knobs (pre-ribosomal processing complexes) are readily recognizable in both contrasted and vitrified samples (Fig. 2). The number of Miller trees in the edited clones appears the same as the number in the parental strain, indicating that no individual ribosomal genes were removed from the genome (Supplementary Fig. 3). This suggests that the integrity of the rDNA repeats as a whole was not impaired by the introduction of multiple double-strand breaks by Cas9.

In the wild-type (WT) *S. cerevisiae* strain IMX672, the apparent intergenic spacer including the *RDN5* gene (2.3 kb) is 500–580 nm long, depending on the route of the DNA and the occupancy of the Pol I-transcribed genes. Chromatin spreads of 5x(O)*RDN5* and 2.5x(J)*HSX1* show a greater distance between Miller trees that can, on the basis of confirming sequencing results, unambiguously be attributed to the incorporated gene repeats (Fig. 2d and Fig. 3a, b, f, g). Specifically, an intergenic spacer that was ~330 nm longer in the clone 5x(O)*RDN5* than in the WT strain proves that three *RDN5* copies were incorporated (one *RDN5* copy is ~110 nm long, depending on the occupancy of the gene by Pol III complexes, Fig. 3b). An increase of ~600 nm in spacer length in the clone 2.5x(J)*HSX1* proves that six *HSX1* copies were incorporated (one *HSX1* copy is ~100 nm long, Fig. 2d). These findings are in agreement with the results of the Southern blot analysis, thus confirming that we can use imaging to precisely read out the number of genes incorporated.

In the parental strain, the endogenous copy of the *RDN5* gene is visible between Miller trees when two or three RNA Pol III complexes are actively transcribing (Fig. 1a). In the edited clones, additional groups of Pol III complexes can be found on the DNA between the trees, at periodic distances from the endogenous gene and from each other, showing a direct correspondence to the inserted copies of the *RDN5* or *HSX1* gene and confirming that

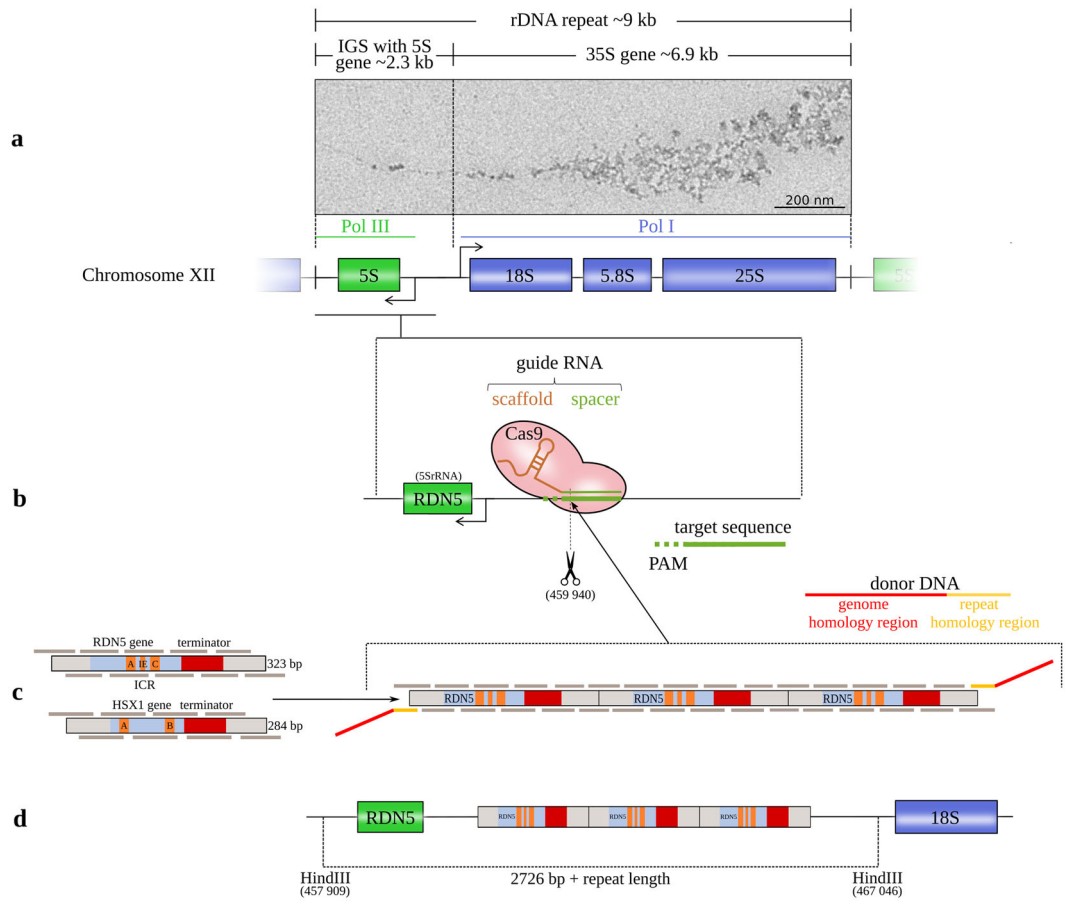

**Fig. 1 Schematic of the method for introducing gene copies. a** The rRNA repeats on *S. cerevisiae* chromosome XII with the 18S, 5.8S, and 25S rRNAs (transcribed by the RNA polymerase I (Pol I)) and the 5S rRNA (transcribed by Pol III in the reverse direction from the complementary strand). The 5S rRNA gene is located in the intergenic spacer (IGS) between successive repeats (Saccharomyces Genome Database standard name: *RDN5-1*). EM image: Miller tree in chromatin spreads of *S. cerevisiae* strain IMX672 at 12,000× magnification from positively stained sample. **b** The CRISPR/Cas9 system induces a double-strand break at the chosen target site with a protospacer adjacent motif (PAM) downstream of the *RDN5* gene (5 S rRNA) within the rRNA repeat. **c** Both copy units cover the gene sequence, the terminator and a 50 bp spacer on both sides. *RDN5* is a type 1 promoter gene with an internal control region (ICR) that contains box A and box C elements and an intermediate element (IE). The internal promoter requires the transcription factors IIIC, IIIB, and IIIA (TFIIIC, TFIIIB, and TFIIIA). *HSX1* is a type 2 promoter gene with box A and box B elements that requires only TFIIIC and TFIIIB[8]. The copy units are divided into overlapping oligomers. The in vivo repeat assembly is targeted to the Cas9 cleavage site by donor DNA adapters. **d** After Cas9 cleavage, the copy cluster is inserted at the DSB site.

these genes were being actively transcribed immediately before chromatin spreading. Clone 5x(O)*RDN5* contains up to 16 additional copies of *RDN5* at each position between Miller trees. The endogenous copy of *RDN5* is visible in all examples as a group of three Pol III complexes close to the start point of the Miller tree (Fig. 3a–c). In Fig. 3a, b, one fully occupied additional *RDN5* gene is visible (as shown by a group of three Pol III complexes). In the region shown in Fig. 3a, this additional *RDN5* gene is directly adjacent to the endogenous gene, while in Fig. 3b, it is separated from the endogenous gene by a DNA stretch whose length suggests the presence of two more incorporated gene copies without associated polymerases. Clone 2.5x(J)*HSX1* contains up to seven copies of *HSX1* in each rDNA repeat. The endogenous copy of *RDN5* is visible, occupied by three or two Pol III complexes (Fig. 2b, c, and d, respectively). In the spread shown in Fig. 2d, three incorporated copies of *HSX1* can be seen, which are contiguous but not directly adjacent to the endogenous *RDN5* gene.

We determined that there was an average of three Pol III complexes present between two Miller trees for the clone 5x(O)*RDN5* (Figs. 3a–c and 4b). As this strain contains an average of 5.3 inserted *RDN5* genes in addition to the endogenous *RDN5*

gene, the maximal number of Pol III complexes transcribing between Miller trees would be 18.9. Thus, 16% of the maximal number of Pol III complexes could be seen, assuming full usage of all *RDN5* genes. Previous reports claim an average of 0.43 Pol III complexes per *RDN5* gene per rDNA repeat in wild-type yeast strains with 143 rDNA repeats[13]. This corresponds to active transcription of only 20–30% of the endogenous genes, each occupied by an average of 1.73 Pol III complexes. We thus achieved a 7-fold increase in the number of Pol III complexes that can be directly analyzed by EM.

The low occupation of the *RDN5* gene in comparison to the Pol I-transcribed rRNA genes has been explained by the accelerated transcription that results from facilitated recycling of Pol III complexes in combination with strict control of transcript numbers by the *RDN5*-exclusive transcription factor III A (TFIIIA)[14]. We used a yeast centromeric plasmid[15] to achieve a mild, constitutive overexpression of TFIIIA in the 5x(O)*RDN5* strain. TFIIIA feedback regulates the amount of 5S rRNA by directly binding it, so an increase in the amount of TFIIIA within the cell should promote the transcription of *RDN5*[14]. In the 5x(O)*RDN5* strain with a constitutive overexpression of TFIIIA, the average number of Pol III complexes between Miller trees was

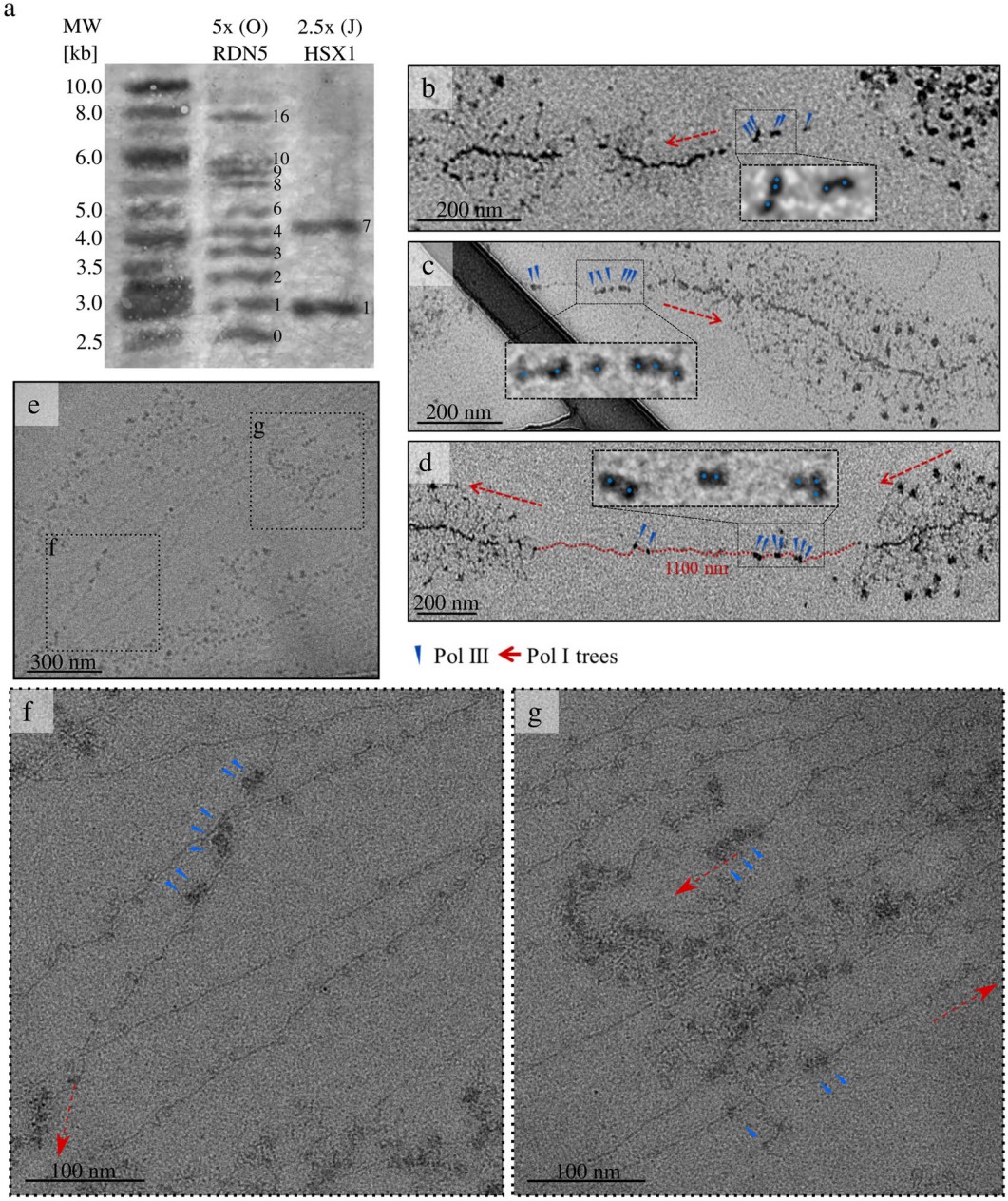

**Fig. 2 Analysis of incorporated gene repeats. a** Southern blot analysis of isolated genomic DNA from selected edited clones. Clones created in wild-type *S. cerevisiae* strain IMX672 background using editing condition 5x(*RDN5*) and 2.5x(*HSX1*) (see Supplementary Table 4). The Southern blot probe labels the genomic fragments with the endogenous copy of the Pol III-transcribed genes along with all inserted copies. The migration height of the band reflects the length of the incorporated copy cluster. After correction for intensity increase due to the higher probe-binding capacities of longer copy clusters, the band intensities can be used to approximate the prevalence of the copy number in relation to other number. We used KPL GeneRuler™ Biotinylated DNA Ladder. The band patterns are annotated with the corresponding copy number. The 5x(O)*RDN5* band labeled '0' derives from the empty repeat incorporation sites (the endogenous *RDN5* copies are themselves part of the rDNA repeats). The clones are denoted with the incorporation condition (see Supplementary Table 4), a consecutive letter and the incorporated gene. **b–d** Miller trees in chromatin spreads of clone 2.5x(J)*HSX1* with digitally enlarged regions showing Pol III complexes. The samples were positively stained with uranyl acetate (UA) and phosphotungstic acid (PTA); EM images were acquired at 12,000× magnification at a defocus of −40 μm. Red dashed arrows label the start or end of a Miller tree and its direction. Red dashed lines show the route of the DNA between Miller trees. **d** The length of the intergenic spacer is indicated in red. Blue arrowheads and blue points label Pol III complexes. **e–g** Miller trees in chromatin spreads of clone 5x(O)*RDN5*. EM images of vitrified sample were acquired at 11,500× magnification at a defocus of −50 μm and with 0.2 e−/Å² total dose (**a**) or 105,000× magnification at a defocus of −3 μm and with 50 e−/Å² total dose (**f, g**). Clone 5x(O)*RDN5* contains up to 16 additional copies of *RDN5* at each position between Miller trees. **f** Cluster of three incorporated *RDN5* units with transcribing Pol III complexes. **g** Endogenous copy of *RDN5* in close proximity to the start of a Miller tree (top) and two incorporated *RDN5* copies with transcribing Pol III complexes (bottom).

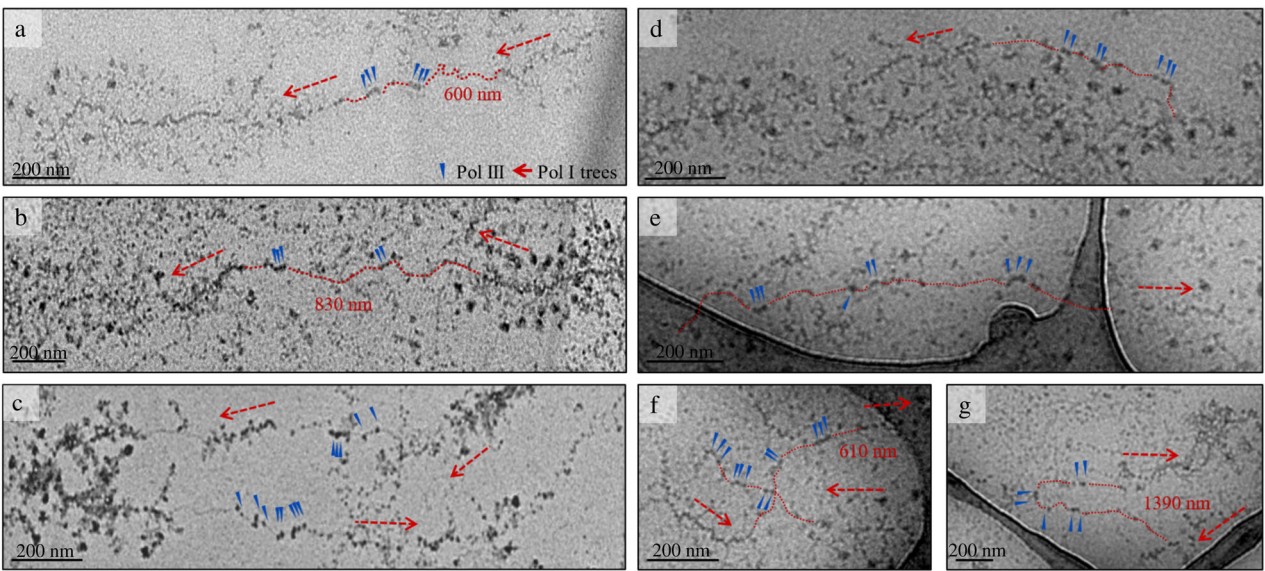

**Fig. 3 Electron microscopic analysis of TFIIIA overexpression in 5x(O)RDN5. a–c** Miller trees in chromatin spreads of the edited clone 5x(O)*RDN5*. Groups of RNA Pol III complexes on endogenous and incorporated *RDN5* genes are visible between Miller trees. **d–g** Spreads of clone 5x(O)*RDN5* with TFIIIA overexpression. The samples were positively stained with UA and PTA; EM images were acquired at 12,000× magnification at a defocus of −40 μm. Red dashed arrows label the start or end of a Miller tree and its direction. Red dashed lines show the route of the DNA between Miller trees. In (**a**, **b**, **f**, and **g**), the length of the intergenic spacer is indicated in red. Blue arrowheads label Pol III complexes.

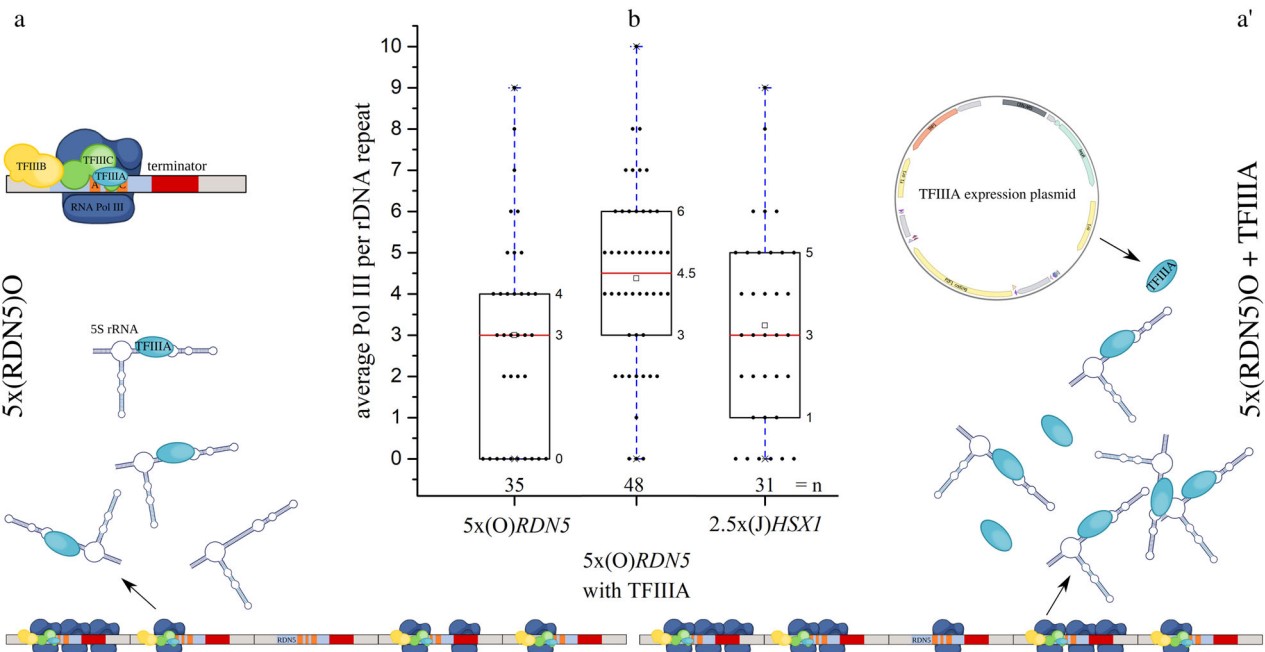

**Fig. 4 Quantitative analysis of TFIIIA overexpression in 5x(O)RDN5. a, a′** Schematic of *RDN5* transcription regulation in strain 5x(O)*RDN5* with and without TFIIIA overexpression. **b** Boxplot analysis of average number of Pol III complexes per rDNA repeat of 5x(O)*RDN5*, 5x(O)*RDN5* with TFIIIA overexpression and 2.5x(J)*HSX1*. Box = 25–75%, bracket = range within 1.5 IQR, red horizontal line = median, square = mean, diamonds = outliers, dots = individual data points. The number of analyzed well-spread Miller tree pairs per strain is indicated below the boxes (Supplementary Table 5). Images of two independent preparations of chromatin for each strain were included in the analysis.

increased to 4.4 in comparison to 3 in the same strain without overexpression (Figs. 3 d–g and 4b). This equals 23% of the maximal number of Pol III complexes for this strain and a 10-fold increase in the number of Pol III complexes between Miller trees in comparison to the wild-type strain. The 46% increase in the number of Pol III complexes upon overexpression of TFIIIA clearly demonstrates that our method for the incorporation and

direct EM visualization of genes can be used practically to observe transcription changes in response to altered cellular conditions.

For clone 2.5x(J)*HSX1*, we determined the average number of Pol III complexes per insertion site to be 3.3 (Figs. 2h–j and 4b). This strain contains an average of 2.3 inserted *HSX1* genes in addition to the endogenous *RDN5* gene. It can harbor a maximal number of 6.7 Pol III complexes between Miller trees, of which

43% could be seen. In relation to the average Pol III presence in unmodified yeast strains, we achieved a 7.7-fold increase. In the case of *HSX1*, the facilitated recycling of the Pol III complex promotes full gene usage at all times, as only one endogenous *HSX1* copy exists in the genome. The *HSX1* gene product, arginine tRNA, is less strictly regulated than the 5S rRNA. No direct feedback regulator like TFIIIA is involved in its transcription and it is thus more resistant to variations in gene copy number and transcription rate (Fig. 1)[8]. This is reflected in our results, which show a higher percentage of active usage of *HSX1* than of *RDN5*. In contrast to *RDN5*, where the gene usage pattern seemed to follow a stochastic distribution, actively transcribed *HSX1* genes preferably appeared in clusters. The *HSX1* gene is not native to the rDNA locus, which can be identified on electron micrographs. Its identification and visualization in complex with the transcribing Pol III complexes would therefore have been impossible in a wild-type strain, as the single gene copy in the genome could not have been distinguished from the surrounding chromatin.

## Discussion

In this work we establish a method for inserting genes between rDNA loci in *S. cerevisiae*, which thereby enriches protein–DNA complexes and enables the ex vivo visualization of macromolecules transcribing, modifying or binding DNA sequences in a native-chromatin environment. At close-to-native conditions, quantitative (cryo-)electron microscopy analysis using Miller trees as landmarks in *S. cerevisiae* chromatin spreads allows for the precise measurement of the number of RNA polymerases and thus evaluation of the transcription rates. We used a specialized CRISPR/Cas9 approach to insert multiple copies of genes of interest near the 35S rRNA gene while preserving the integrity of the rDNA repeat as a whole, and achieving sufficient stability to enable repeated electron microscopic analysis of the same strain. We demonstrated successful genomic integration of multiple copies of the functional *RDN5* and *HSX1* genes and showed that our approach is able to enrich RNA Pol III at these sites. In the case of the locus-foreign *HSX1*, the localization of the gene itself and its transcription complexes with Pol III would have been impossible without using the method presented here. We also demonstrated that changes in transcriptional activity can be observed on the incorporated genes upon overexpression of a transcription factor. However, our method may alter the genes' transcriptional status by placement of genes within the euchromatin environment of the rDNA repeat and multiplication of the gene copy number. The transcriptional activity obtained by our method allows for comparison within the same strain under different experimental conditions, although, it cannot directly be compared to data obtained by other methods. The design of our method primarily aims toward the structural elucidation of complexes between specific genes and polymerases, transcription factors, or modifying enzymes. However, the number of binding events observed cannot be used to quantify the state of the single-copy gene at its native locus within the genome.

Here we studied Pol III-transcribed genes. The Pol III complex is a native component of transcription events within the nucleolus, even though many of the Pol III-transcribed genes are not—or not permanently—associated with the nucleolus[16]. In the future it can be studied whether genes transcribed by the RNA polymerase II (Pol II), which is localized in the nucleoplasma[17], experience different activities. While stalled Pol II contributes to the nucleolar architecture by binding non-coding rDNA spacers[18], it is unclear whether the polymerase would deploy regular transcriptional activity on genes placed within the nucleolus, also given its need for numerous different transcription

factors that might not be native to the nucleolus. In summary, we describe a method to increase the yield of actively transcribing Pol III complexes that are identifiable on electron micrographs of spread chromatin, as well as a tool for ex vivo investigations of gene transcription, modification like acetylation and methylation, or binding by transcription factors that could be generally applicable to many sequences of interest.

## Methods

**Strains, plasmids, primers, and oligonucleotides.** The Cas9-expressing *S. cerevisiae* strain IMX672 (Euroscarf, Oberursel, Germany, #Y40595[9]) was used as a parental strain for all primary incorporation experiments. Strains were grown at 30 °C and 200 rpm in YPD-2% glucose medium or in synthetic complete drop-out medium (SC medium) without uracil supplement for the maintenance of KlURA3-bearing (URA3 gene from *Kluyveromyces lactis*) plasmids. The oligonucleotides assembling the repeat were designed as recommended by Lancrey et al.[12] with a length of 50–60 nucleotides and an overlap of 20 nucleotides (for one oligonucleotide, an overlap of 28 nucleotides was required). All oligonucleotides and primers used in this study were purchased from Metabion (Planegg, Germany) and are listed in Supplementary Tables 1 and 3. Plasmids were purchased from Addgene (Watertown, MA, USA).

**Design of CRISPR/Cas9 double-strand induction.** A target sequence 100 bp downstream of the *RDN5* terminator was chosen with the CRISPOR tool[19]. The respective guide sequence and protospacer adjacent motif (PAM) on the reverse strand were TAGCAACCCAATGAGCATAA - TGG. The guide RNA was delivered into the cells via the plasmid pMEL10 (Addgene, #107916)[9]. pMEL10 was linearized by PCR with the primers p_1.1_fw and p_1.1_rev using the Phusion HF DNA polymerase (New England Biolabs). We designed a double-stranded genomic DNA cassette comprising the 20 bp guide sequence in between two 50 bp homology arms, matching the plasmid backbone on both sides of the guide sequence site. The insertion of the guide sequence into the vector backbone was facilitated by homologous recombination in the *E. coli* strain DH5α. Correct insertion was verified by Sanger sequencing. Following Lancrey et al.[12], donor DNAs were composed of 32–57 bp of the reverse beginning or forward end of the repeat sequence and 200 bp or 808–833 bp of reverse or forward genomic DNA left or right of the Cas9 restriction site. The donor DNAs were synthesized by PCR with isolated genomic DNA of the *S. cerevisiae* strain YPH499 (MATa ura3-52 lys2-801_amber ade2-101_ochre trp1-Δ63 his3-Δ200 leu2-Δ1) or IMX672 (MATa ura3-52 trp1-289 leu2-3,112 his3D can1D::cas9-natNT2) as the template, using the Phusion HF DNA polymerase. PCR was conducted with the primer pairs p_3.1_fw/p_3.2_rev and p_3.3_fw/p_3.4_rev for 200 bp homology arms with the genomic regions flanking the copy insertion site and primer pairs p_3.5_fw/p_3.6_rev (*RDN5*), p_3.7_fw/p_3.8_rev (*RDN5*), p_3.9_fw/p_3.6_rev (*HSX1*), and p_3.10_fw/ p_3.8_rev (*HSX1*) for >800 bp homology arms. The sequences of the left and right donor DNA pieces are presented in Supplementary Table 2.

**In vivo repeat assembly and genomic incorporation.** The *S. cerevisiae* strain IMX672 was transformed with pMEL10_gRNA, the repeat-forming oligonucleotides and both donor DNAs (either short or long versions) following the protocol reported by Gietz and Schiestl[20]. Various amounts of each component were tested (Supplementary Table 4). The maximal amounts of the components were limited by the DNA uptake capabilities of the yeast cells. pMEL10-gRNA-positive cells were selected via the KlURA3 auxotrophy marker.

**Verification of repeat insertion by Southern blotting.** Southern blotting followed the protocol by Southern[21]. Genomic DNA of 6–51 yeast transformants per transformation condition was isolated according to the protocol by Hanna and Xiao[22] and digested with HindIII (New England Biolabs), leading to a fragmentation of the genomic DNA but not the inserted repeats. The fragments were used for Southern blots with biotin-labeled probes synthesized by PCR from customized plasmids carrying the repeat sequences of the *RDN5* or *HSX1* gene (Eurofins Genomics) with the Biotin PCR Labeling Core Kit (Jena Bioscience) and primers p_5.1_fw and p_5.1_rev. Both probes covered the entire repeat units. The Southern blot probe labels the genomic fragments with the endogenous copy of the Pol III-transcribed genes along with all inserted repeats. However, the endogenous *HSX1* gene is locus-foreign and does not appear on the Southern blot as the digestion of the genome was optimized to yield a well-resolvable fragments length of the incorporation site only. The migration height of the band reflects the length of the incorporated repeat. After correction for intensity increase due to the higher probe-binding capacities of longer repeats, the band intensities can be used to approximate the prevalence of the repeat length in relation to other lengths. For fragment size approximation, we used KPL GeneRuler™ Biotinylated DNA Ladder (ThermoFisher).

**Verification of repeat insertion by sequencing.** To sequence the inserted gene repeats, the genomic DNA of the strains 5x(O)*RDN5* and 2.5x(J)*HSX1* was isolated.

The inserted repeats were then amplified with one primer positioned upstream of the insertion site and A-tailed with terminal transferase. The complementary strain was synthesized with the help of a poly-T primer. Remaining genomic DNA was removed by PEG precipitation and the resulting library was blunt-end cloned and Sanger sequenced. The strains 5x(O)RDN5 and 2.5x(J)HSX1 are available on request from the corresponding author.

**Overexpression of TFIIIA.** The plasmid p414-TEF1p-Cas9-CYC1t (DiCarlo et al.[15], Addgene, #43802) was linearized with the primer pair p_6.1_fw/p_6.2_rev to yield the backbone without the Cas9 gene. The *PZF1* gene, coding for TFIIIA, was PCR-amplified from the genome of IMX672 with the primer pair p_6.3_fw/p_6.4_rev, containing 40 bp homology arms to the backbone. The insertion of *PZF1* into the vector backbone was facilitated by homologous recombination in the strain 5x(O)RDN5, and positive clones were selected via the TRP1 auxotrophy marker. Correct insertion was verified by Sanger sequencing. The strain 5x(O)RDN5 with p414-TEF1p-PZF1-CYC1t is available on request from the corresponding author.

**Spreading of yeast chromatin.** The spreading of yeast chromatin on EM grids was based on the protocol by Osheim et al.[23] with several optimizations that were developed by our group. All solutions used in the spreading procedure were prepared in double-distilled $H_2O$ and filtered through a thoroughly pre-rinsed bottle-top filter with pore size 0.22 μm to remove impurities. On the day of each spreading procedure, the pH of each solution was adjusted with pH10 buffer (24.53 mM $Na_2CO_3$, 30.94 mM $NaHCO_3$, pH 10). The pH of paraformaldehyde-based fixation solutions was pre-adjusted to 8.3 by stirring with mixed-bed resin (#M8032, Sigma-Aldrich) to reduce the amount of required pH10 buffer. Edited yeast strains were grown in YPD medium with 1 M sorbitol. Then 1–3 mL from cultures at an OD600 of 0.2–0.4 were harvested, washed in medium, collected by centrifugation, resuspended in 1 mL medium, and added to 200 μL prewarmed zymolyase working stock (25 mg/mL zymolase-20T in 0.1 M $K_2HPO_4$, 0.1 M $KH_2PO_4$, 0.5 mM $MgCl_2$, 1.2 M D-sorbitol, pH 6.2). Cell walls were partially digested by incubation for 5 min at 30 °C at 600 rpm on an Eppendorf Thermomixer, during which the cell suspension was shortly vortexed at full speed after each minute. Cells were spun at 16,000g for 15–30 s. Medium was removed immediately by pipetting and wiping with Kimtech wipes. Subsequently, 1 mL of lysis buffer (0.025% (v/v) Triton-X 100, pH 9.25) was added, while slowly pipetting up and down. The resulting suspension was added to 6 mL spreading solution (12.5 mM KCl, pH 9) in a 100 mL Erlenmeyer flask, and lysis was supported by swirling the flask and pipetting up and down. Then, 1 mL of the lysate was added to a 35 mm PS petri dish that was incubated on a rocking table for 45 min. Chromatin and associated proteins were fixed in 1% formaldehyde (Science Services, EM grade) and 10 mM sucrose for 15 min at room temperature. Chromatin was deposited on EM grids in custom centrifugation chambers designed by our group, which were inserted into 50 mL conical tubes. Before deposition, 200 mesh Lacey carbon grids coated with a 4–6 nm carbon film were glow discharged. Grid chambers were filled with fixation solution (10% (v/v) formaldehyde, 0.1 M sucrose, pH 8.8) before dropping grids to the bottom of the chamber. After grid insertion, half of the respective solution was removed, leaving a cushion of fixation solution. Lysate was added on top of the cushion in the chamber, creating a slightly convex surface. Chambers were sealed using a glass cover slip. Chromatin was deposited on the carbon film by centrifugation at 1800g and 4 °C for 8 min with acceleration and deceleration set to zero. Grids were retrieved from grid chambers, washed in excess wash buffer (0.06% (v/v) Kodak Photo-Flo 200, pH 8.8) for 30 s, and air-dried.

**Positive staining.** The contrast of chromatin and associated proteins was enhanced by positive staining with uranyl acetate (UA) and phosphotungstic acid (PTA) in 96-well plates. For each grid, one well with 1% PTA in 75% ethanol, one well with 1% UA in MiliQ-$H_2O$, and 8 wells with 100% ethanol were prepared. Grids were submerged in the PTA well for 30 s and then immediately sequentially washed in four ethanol wells. The remaining PTA and ethanol were removed from the forceps by blotting with filter paper, and grids were immediately submerged in 1% UA for 1 min, before a second round of sequential washing and air-drying. Grids were stored in a chamber with reduced humidity.

**Electron microscopy.** Positively stained samples were examined in a Tecnai F30 G[2] transmission electron microscope (Thermo Fisher Scientific) at 300 kV. Electron micrographs at 12,000× or 15,500× magnification were recorded with the Thermo Fisher Scientific software TIA (TEM Imaging & Analysis version 4.15 SP1) or EPU (version 1.11.1.50REL) on a Falcon 3EC camera (Thermo Fisher Scientific) operated in linear mode. High-contrast images were recorded with ~50 e⁻/pixel total dose and a defocus of −30 to −70 μm.

Vitrified samples were examined in a Titan Krios transmission electron microscope (FEI) at 300 kV with a GATAN GIF Quantum post-column energy filter in zero-loss peak mode. Electron micrographs of lysed yeast cells at 11,500× magnification were recorded with the software SerialEM v3.8.0beta[24] on a K2 Summit direct detector (Gatan). Screening images were acquired with a dose of 0.05 e⁻/Å² and images of spread chromatin with a dose of 0.2 e⁻/Å² (dose-fractionated) and a defocus of −50 μm. Images of regions of interest within

spread chromatin were acquired at 105,000× magnification in super-resolution mode (pixel size 0.65 Å²) with a dose of 50 e⁻/Å² (dose-fractionated) and a defocus of −3 μm.

**Statistics and reproducibility.** The genomic stability of the clones (general reproducibility) was confirmed by Southern blot analysis before and after repetitive re-plating (> ten rounds over a time course of >6 month) of the clones 5x(O)RDN5 and 2.5x(J)HSX1 (Fig. 2 and Supplementary Fig. 2c). The analysis after propagation was done with distinct triplicates. Growth curves of IMX672 (WT), 5x(O)RDN5, and 2.5x(J)HSX1 (Supplementary Fig. 4) were done for three distinct replicates per strain. For the box blot analysis of maximal inserted copy numbers per strain in relation to the different experimental incorporation conditions (Supplementary Fig. 5), sample sizes were based on the maximal number of samples that could be analyzed in parallel by Southern blotting. No data were excluded, all analyzed clones were included in the box plots. For the quantitative analysis of the average number of Pol III complexes per rDNA repeat of 5x(O)RDN5, 5x(O)RDN5 with TFIIIA overexpression, and 2.5x(J)HSX1 (Fig. 4 and Supplementary Table 5), the sample sizes were chosen to be >30 to allow proper comparability in a box plot and limited by the acquisition capacities of two microscope sessions. Images of two independent preparations of chromatin based on two different plates were included in the analysis to ensure reproducibility, and all acquired images with well-spread Miller trees were included in the analysis.

**Reporting summary.** Further information on research design is available in the Nature Research Reporting Summary linked to this article.

## Data availability
The data that support the findings of this study are available on request from the corresponding author. These include raw Southern blot membranes and raw electron micrographs.

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

## Acknowledgements

This work was funded by the Deutsche Forschungsgemeinschaft grants FR1653/12 and FR1653/14 as well as SFB902 (project B5). We thank our colleagues Victor-Valentin Hodirnau and Julian Reitz for developments in the DNA spreading and conceptual insights.

## Author contributions

S.M., U.H.E., and A.S.F. conceived and designed the experiments, S.M. performed the experiments and analyzed the data, and S.M. and A.S.F. wrote the paper.

## Funding

## Competing interests

The authors declare no competing interests.
