## [Peer Review File · Communications Biology]

Reviewers' comments:

Reviewer #1 (Remarks to the Author):

In the submitted manuscript, the authors reported the use of CRISPR/Cas9 to stably insert multiple copies of a particular gene-of-interest near the 35S rRNA gene, so that the gene-of-interest can be more readily localized and imaged in the electron microscope. As a proof-of-concept, the authors inserted additional copies of either the locus-native gene RDN5 or the locus-foreign gene HSX1 and showed an enrichment of Pol III complexes associated with these genes. They also demonstrated that when they overexpressed TFIIIA, they could visually detect an increase in the number of Pol III complexes. Overall, I find the presented biophysical method to be interesting. Below are some comments that I have for improving the manuscript.

- 1) The authors did not mention the possibility of a position effect in their manuscript. For example, a gene that is silenced in heterochromatin may become actively transcribed when inserted near the 35S rRNA gene. This will not be physiologically relevant. I think the authors can discuss more extensively the use cases and caveats of their method.
- 2) The authors can evaluate the genome stability of their modified strains more rigorously. Three rounds of replating is too short a time for anyone to definitively conclude that the tandem array is stable.
- 3) The authors wrote "This massive manipulation of genomic integrity impaired neither cell growth nor the transcription of the rDNA repeats ..." Please show the data.

Reviewer #2 (Remarks to the Author):

Brief summary of the manuscript

In this manuscript, Manger et al describe a method to visualize specific gene activity by electron microscopy in yeast. First, the authors successfully inserted repeats of a gene of interest in the yeast genome at the rDNA locus (between RDN5 and RDN18 in chromosome XII), using a recently devised genome editing approach for building repeated DNA in yeast. Using this technique, they obtain strains with multiple copy of the inserted gene, which they prove by Southern blotting. They also provide sequencing evidence for correct integration and faithful gene assembly at the desired locus.

The two inserted genes are the RDN5 gene, native to the locus where insertions are performed, and HSX1, which codes for a tRNA. Both genes are transcribed by polIII and are therefore allowed for transcription in the nucleolus, where the rDNA locus resides. Then, the authors use electron microscopy to visualize yeast chromatin spreads, where the rDNA locus forms a recognizable structure known as Miller trees, and where the site of polIII transcribed RDN5 is visible in between Miller trees. In the engineered strains with multiple inserted genes, more than one cluster of polIII can be observed, indicating that the inserted genes are functionally transcribed by polIII. This result is true with the native gene RDN5 and also for another polIII transcribed gene HSX1. Interestingly, the authors show a small increase in the number of polIII clusters in average in a strain overexpressing TFIIIA.

Overall impression of the work

The study combines in a clever way the characteristics of rDNA's aspect under electronic microscopy and up to date genomic edition techniques. Overall, the article is well written and the experiments presented support convincingly the authors claim. I have however several minor comments that should be addressed by the authors.

Specific comments

1. The title is too general. This study is an interesting proof of concept experiment for polIII genes at a specific locus (the rDNA), and the title should reflect this specificity. It is likely that this approach is not applicable to polII transcribed genes, or at least this limitation should be addressed. Since the authors only show examples with polIII genes, the title should reflect this fact.

2. The first sentence reads "...epigenetic events, such as transcription, modification or binding". Which modifications or binding are the authors referring to? The sentence is too vague in the context.

3. The authors show a small increase in polIII occupancy upon overexpression of the transcription factor TFIIIA (Figure 4). Although the data supports the authors' claim that the system can be used to observe change in transcription, the authors do not show the effect of TFIIIA on the ectopically inserted HSX1 genes. For this experimental approach to be useful for observing specific genes in an ectopic context by electron microscopy, it would have been interesting to show that TFIIIA overexpression affects HSX1 as well.

4. In figure A2, Southern blots are shown without scale. The authors should provide a DNA ladder next to the blots. Also, figures B and C do not prove that the inserted repeats are stable. As such, the data show that clones with multiple insertions can be propagated. The authors claim that the higher bands visible on the Southern blot of the propagated strain's genomic DNA are due to increased transfer efficiency, but they could be due to an early recombination event during propagation or different DNA digestion efficiency during DNA preparation. The authors should refrain from making claims about strain stability without careful quantitative assessment of copy number using fluctuation assays.

5. The authors present in table A4 the composition of the various oligonucleotide and gRNA mixes that they used to obtain multiple inserted yeast strains. In the text, this table is referred to as a "statistical analysis in relation to the different experimental incorporation conditions".

The term statistical analysis is improperly used here.

The authors however should report how many copies were inserted in the 10X clone. It would have been interesting, in regard to assessing the efficiency of the genome editing method, to see if there was a correlation, if any, between the oligonucleotide ratios used in table A4 and the number of inserted genes.

We would like to thank the referees for the very constructive suggestions.

Referee #1

Comment	Answer	Updated text
1) The authors did not mention the possibility of a position effect in their manuscript. For example, a gene that is silenced in heterochromatin may become actively transcribed when inserted near the 35S rRNA gene. This will not be physiologically relevant. I think the authors can discuss more extensively the use cases and caveats of their method.	Answer: Thank you for raising this important point. We strongly extended our discussion on the point you raised concerning the placement of the genes within the nucleolar euchromatin, along with limitations of the method deriving from the multiplication of the gene number and the possible impracticality on Pol II-transcribed genes (L. 282-290).	Discussion, L.272-281 We also demonstrated that changes in transcriptional activity can be observed on the incorporated genes upon overexpression of a transcription factor. However, our method may alter the genes transcriptional status by placement of genes within the euchromatin environment of the rDNA repeat and multiplication of the gene copy number. The transcriptional activity obtained by our method allows for comparison within the same strain under different experimental conditions, although, it cannot directly be compared to data obtained by other methods. The design of our method primarily aims towards the structural elucidation of complexes between specific genes and polymerases, transcription factors or modifying enzymes. However, the number of binding events observed cannot be used to quantify the state of the single-copy gene at its native locus within the genome.
2) The authors can evaluate the genome stability of their modified strains more rigorously. Three rounds of replating is too short a time for anyone to definitively conclude that the tandem array is stable.	Answer: We have updated the manuscript with respect to claims of stable insertion. The main goal of our method is to generate strains that allow EM analysis of short and lowly abundant DNA stretches and interacting proteins. As such, complete long-term stability is not strictly required, especially since the edited strains can be cryo-preserved. Nevertheless, we repeated the Southern blot analysis of the two strains used here after > 10 rounds of re-	L. 107-109: We confirmed the six-month stability of the genomic DNA by repeated Southern blot analysis of two edited clones after more than ten rounds of re-plating (Supplementary Figure 2c). L. 265-268: We used a specialized CRISPR/Cas9 approach to insert multiple copies of genes of interest near the 35S rRNA gene while preserving the integrity of the rDNA repeat as a whole, and achieving sufficient stability to

	plating over a course of > 6 month in triplicates from individual plates. The new SB can be found in Supplementary Figure 2c (see below). In our opinion, this result, combined with the repeated visual analysis of these strains using electron microscopy, demonstrates sufficient stability of the integrated repeats to achieve the main goal of our method.	enable repeated electron microscopic analysis of the same strain.
3) The authors wrote "This massive manipulation of genomic integrity impaired neither cell growth nor the transcription of the rDNA repeats ..." Please show the data.	Answer: We now included growth curves in triplicates of the wild type along with the two used modified strains as Supplementary Figure 4 (see below) to support our claim that the growth rates are not (strongly) affected by the genome engineering. We also provided overviews of the rDNA region that give an impression of the total number of Miller trees and their overall appearance across strains as Supplementary Figure 3 (see below).	

Referee #2

Comment	Answer	Updated text
1. The title is too general. This study is an interesting proof of concept experiment for polIII genes at a specific locus (the rDNA), and the title should reflect this specificity. It is likely that this approach is not applicable to polII transcribed genes, or at least this limitation should be addressed. Since the authors only show examples with polIII genes, the title should reflect this fact.	Answer: Thank you for raising this important point. We changed the title and several paragraphs of the manuscript to make clear that the observed genes are Pol III-transcribed and placed within the nucleolus. We also strongly extended our discussion on the point you raised concerning Pol II-transcribed genes, along with limitations of the method deriving from placement of the genes within the nucleolar euchromatin and the multiplication of the gene number (L.273-281).	Title: Ex vivo visualization of RNA polymerase III-specific gene activity with electron microscopy L. 28-29: (...) near the 35S rRNA gene, which is a frequently occurring, easy-to-identify genomic locus within the nucleolus that can be used as a landmark in electron micrographs. L. 57-58: (...) in a native chromatin environment within the nucleolus. Discussion, L.282-290: Here we studied Pol III-transcribed genes. The Pol III complex is a native component of

		transcription events within the nucleolus, even though many of the Pol III-transcribed genes are not – or not permanently – associated with the nucleolus [16]. In the future it can be studied whether genes transcribed by the RNA polymerase II (Pol II), which is localized in the nucleoplasm [17], experience different activities. While stalled Pol II contributes to the nucleolar architecture by binding non-coding rDNA spacers [18], it is unclear whether the polymerase would deploy regular transcriptional activity on genes placed within the nucleolus, also given its need for numerous different transcription factors that might not be native to the nucleolus.
2. The first sentence reads “...epigenetic events, such as transcription, modification or binding”. Which modifications or binding are the authors referring to? The sentence is too vague in the context.	Answer: The first sentence of the abstract was meant to give an impression about different kinds of events that could generally be observed by direct imaging of individual genes at molecular resolution. Modification events can be acetylation or methylation, binding events can be attachment or detachment of transcription factors to or from the DNA. As we only observed transcription and transcription factor binding, we changed the first sentence and parts of the discussion accordingly (L.240-241). While our proof-of-principle study focused on gene transcription by RNA Pol III, which can readily be observed at low magnifications based on its size, larger datasets at higher magnifications on the RDN5 and HSX1 genes in our modified strains would also reveal the transcription factors bound to the internal promoter regions.	Abstract, L. 21-22: The direct study of transcription or DNA-protein-binding events, requires imaging of individual genes at molecular resolution. Discussion, L. 290-294: In summary, we describe a method to increase the yield of actively transcribing Pol III complexes that are identifiable on electron micrographs of spread chromatin, as well as a tool for ex vivo investigations of gene transcription, modification like acetylation and methylation, or binding by transcription factors that could be generally applicable to many sequences of interest.

3. The authors show a small increase in polIII occupancy upon overexpression of the transcription factor TFIIIA (Figure 4). Although the data supports the authors' claim that the system can be used to observe change in transcription, the authors do not show the effect of TFIIIA on the ectopically inserted HSX1 genes. For this experimental approach to be useful for observing specific genes in an ectopic context by electron microscopy, it would have been interesting to show that TFIIIA overexpression affects HSX1 as well.	Answer: An effect of the overexpression of TFIIIA on HSX1 cannot be expected as the transcription factor is RDN5-exclusive. In Figure 1 we attempted to show that RDN5 is a type 1 promotor gene with internal box A and box C elements and an intermediate element (IE). The internal promotor requires the transcription factors TFIIIC, TFIIIB and TFIIIA. HSX1, on the other hand, is a type 2 promotor gene with box A and box B elements that requires only TFIIIC and TFIIIB. We slightly modified the text in the respective result section to make this clearer (L. 196-198).	L. 238-241: The HSX1 gene product, arginine tRNA, is less strictly regulated than the 5S rRNA. No direct feedback regulator like TFIIIA is involved in its transcription and it is thus more resistant to variations in gene copy number and transcription rate (see Figure 1) [8].
4a. In figure A2, Southern blots are shown without scale. The authors should provide a DNA ladder next to the blots.	Answer: These blots were used purely quantitatively to pre-select strains and done with a non-biotin-labelled DNA ladder so that the bands were very faint and not properly visible on the digital scans of the blots. Nevertheless, the strains used for the following experiments were re-analysed in more detail and with a biotin-labelled DNA ladder (see Figure 2 in main manuscript and Supplementary Figure 2c). Supplementary Figure 2a is only shown here to give the reader an impression about the general outcome (incorporated gene copies) of the incorporation experiments.	
4b. Also, figures B and C do not prove that the inserted repeats are stable. As such, the data show that clones with multiple insertions can be propagated. The authors claim that the higher bands visible on the Southern blot of the propagated strain's genomic DNA are due to increased	Answer: We have updated the manuscript with respect to claims of stable insertion. The main goal of our method is to generate strains that allow EM analysis of short and lowly abundant DNA stretches and interacting proteins. As such, complete long-term stability is	L. 107-109: We confirmed the six-month stability of the genomic DNA by repeated Southern blot analysis of two edited clones after more than ten rounds of re-plating (Supplementary Figure 2c). L. 265-268: We used a specialized CRISPR/Cas9 approach to insert multiple

transfer efficiency, but they could be due to an early recombination event during propagation or different DNA digestion efficiency during DNA preparation. The authors should refrain from making claims about strain stability without careful quantitative assessment of copy number using fluctuation assays.	not strictly required, especially since the edited strains can be cryo-preserved. Nevertheless, we repeated the Southern blot analysis of the two strains used here after > 10 rounds of re-plating over a course of > 6 month in triplicates from individual plates. The new SB can be found in Figure A2C (see below). In our opinion, this result, combined with the repeated visual analysis of these strains using electron microscopy, demonstrates sufficient stability of the integrated repeats to achieve the main goal of our method.	copies of genes of interest near the 35S rRNA gene while preserving the integrity of the rDNA repeat as a whole, and achieving sufficient stability to enable repeated electron microscopic analysis of the same strain. Supplementary Figure 2, figure description: The band pattern of 2.5x(J)HSX1 perfectly matches that on the SB in Figure 2 (clone from original plate). The band pattern of 5x(O)RDN5 matches that in Supplementary Figure 2b (after three rounds of re-plating) and Figure 2 (after ~ five rounds of re-plating). In comparison to the band pattern in Supplementary Figure 2a (clone from original plate), additional, higher-migrating bands are visible at the top of the pattern. These could derive from a lower DNA digestion or DNA transfer efficiency of the earliest SB or an early recombination event during strain propagation. As no further changes in band pattern occurred between more-often re-plated strains in later SB analyses, we consider the first explanation to be more likely. Later SB analyses (Supplementary Figure 2c and Figure 2) have been performed with digestion and transfer times of > 16h to promote appearance of higher-migrating bands.
5. The authors present in table A4 the composition of the various oligonucleotide and gRNA mixes that they used to obtain multiple inserted yeast strains. In the text, this table is referred to as a “statistical analysis in relation to the different experimental incorporation conditions”. The term statistical analysis is	Answer: This is probably a misunderstanding based on unfortunate numbering of the figures and tables. In Table A4 we provide the amounts of components used to transform IMX672 for CRISPR/Cas9 incorporation experiments (different experimental incorporation conditions. The statistical analysis of the	

improperly used here. The authors however should report how many copies were inserted in the 10X clone. It would have been interesting, in regard to assessing the efficiency of the genome editing method, to see if there was a correlation, if any, between the oligonucleotide ratios used in table A4 and the number of inserted genes.

maximal incorporated repeat copy numbers in relation to the different experimental incorporation conditions are presented in Figure A5 (Boxplot) - formerly Figure A4 before additional figures were included in the supplements. This figure shows that the number of incorporated gene copies was highest in the 5x condition and decreased again in the 10x condition for both genes (see below).

Supplementary Figure 2: Southern blot (SB) analysis of isolated genomic DNA. Dashed arrows mark the bands of the endogenous *RDN5* gene copies and the empty incorporation sites (the endogenous copies are themselves part of the rDNA repeats). **(a)** Exemplary SB analysis of clones created in IMX672 (WT, see dotted box) background using editing condition 5x(*RDN5*) (see Supplementary Table 4). The clone inside the dashed box (5x(O)*RDN5*) was used for further experiments. **(b)** SB analysis of iteratively edited and propagated clones. Clones were created in 5x(O)*RDN5* background using editing condition 5x (see Supplementary Table 4). The clone inside the dashed box (5x(O)*RDN5*) was re-analyzed after three rounds of re-plating (*). **(c)** SB analysis of the long-term stability of the clones 2.5x(J)*HSX1* and 5x(O)*RDN5*. Both clones were re-analyzed after > ten rounds of re-plating over the course of > six month (**) in triplicates from individual plates. We used KPL GeneRuler™ Biotinylatd DNA Ladder. The band pattern of 2.5x(J)*HSX1* perfectly matches that on the SB in Figure 2 (clone

from original plate). The band pattern of 5x(O)*RDN5* matches that in Supplementary Figure 2b (after three rounds of re-plating) and Figure 2 (after ~ five rounds of re-plating). In comparison to the band pattern in Supplementary Figure 2a (clone from original plate), additional, higher-migrating bands are visible at the top of the pattern. These could derive from a lower DNA digestion or DNA transfer efficiency of the earliest SB or an early recombination event during strain propagation. As no further changes in band pattern occurred between more-often re-plated strains in later SB analyses, we consider the first explanation to be more likely. Later SB analyses (Supplementary Figure 2c and Figure 2) have been performed with digestion and transfer times of > 16h to promote appearance of higher-migrating bands.

Supplementary Figure 3: Overview of Miller trees in chromatin spreads of (a) IMX672 (WT), (b) 2.5x(J)*HSX1* and (c) 5x(O)*RDN5*. The rDNA repeat seems unchanged in overall appearance and numbers of the Miller trees. The samples were positively stained with UA and PTA; EM images were acquired at 12,000x magnification at a defocus of $-40\ \mu\text{m}$.

Supplementary Figure 4: Growth of IMX672 (WT), 5x(O)*RDN5* and 2.5x(J)*HSX1*. The OD_{600} of the strains were measured every hour over the course of 18 hours for three distinct replicates per strain. The curves were analyzed by logarithmic transformation and linear fit from $t = 3$ to $t = 13$ (exponential growth phase) with line slopes being 0.189 ± 0.006 ($R^2 = 0.992$) for IMX672 (WT), 0.185 ± 0.004 ($R^2 = 0.996$) for 5x(O)*RDN5* and 0.179 ± 0.006 ($R^2 = 0.994$) for 2.5x(J)*HSX1*. Error bars represent standard deviation.

Supplementary Figure 5: Boxplot analysis of maximal incorporated copy numbers for all experimental

conditions. Box = 25–75%, bracket = range within 1.5 IQR, horizontal line = median, square = mean, diamonds = outliers. Boxplots include only clones with a positive incorporation result. The number of positive clones and their percentage in relation to all analyzed clones are indicated below the boxes. Sample-sizes were based on the maximal number of samples that could be analyzed in parallel by Southern Blotting. No data was excluded, all analyzed clones were included in the box plot.

REVIEWERS' COMMENTS:

Reviewer #1 (Remarks to the Author):

The authors submitted a revised manuscript of their work on using CRISPR to insert multiple copies of a gene-of-interest near the 35S rRNA gene to enable its localization and visualization by EM. Overall, I think that the authors have addressed my previous comments adequately and hence, I am satisfied with the revised manuscript.

Reviewer #2 (Remarks to the Author):

In this revised version, the reviewer acknowledges that his comments and concerns expressed in the first review were all taken into accounts and properly addressed. I have no more concerns regarding the interest of the proposed work for publication in communications biology.